# Enhancing Graph Tasks with a Dual-Block Graph Transformer: A Synergistic Approach to Local and Global Attention

## Abstract

In this work, we address the limitations of traditional Transformers in graph tasks. While some approaches predominantly leverage local attention mechanisms akin to Graph Neural Networks (GNNs), often neglecting the global attention capabilities inherent in the Transformer model. Conversely, other methods overly focus on the global attention aspect of the Transformer, ignoring the importance of local attention mechanisms in the context of graph structure. To this end, we propose a novel Message Passing Transformer with strategic modifications to the original Transformer, significantly enhancing its performance on graph tasks by improving the handling of local attention. Building on this, we further propose a novel Dual-Block Graph Transformer that synergistically integrates local and global attention mechanisms. This architecture comprises two distinct blocks inside each head: the Message Passing Block, designed to emulate local attention, and a second block that encapsulates the global attention mechanism, mirroring the original Transformer model. This dual-block design inside each head enables our model to capture both fine-grained local and high-level global interactions in graph tasks, leading to a more comprehensive and robust graph representation. We empirically validate our model on node classification tasks, particularly on heterophilic graphs, and graph classification tasks. The results demonstrate that our Dual-Block Graph Transformer significantly outperforms both GNN and Graph Transformer models. Remarkably, this superior performance is achieved without the necessity for complex positional encoding strategies, underscoring the efficacy of our approach.

## 1 Introduction

The Transformer model, introduced by Vaswani et al. (2017), has revolutionized the field of natural language processing with its innovative attention mechanism, known as the self-attention or global attention. This mechanism allows the model to weigh the importance of different elements in a sequence, thereby capturing long-range dependencies and complex patterns. Despite its success in sequence-based tasks, the application of the Transformer model to graph-based tasks remains challenging due to the inherent differences between sequences and graphs.

Graph Transformers, an extension of the traditional Transformer model, have been proposed to address the challenges of processing graph-structured data (Battaglia et al., 2018). These models adapt the Transformer's self-attention mechanism to handle graph data, enabling them to capture complex patterns in the graph structure (Rong et al., 2021). By doing so, Graph Transformers can effectively model the dependencies between nodes in a graph, which is crucial for many tasks such as node classification, link prediction, and graph classification (Zhang & Chen, 2018; Wu et al., 2019).

Traditional Graph Transformer models are primarily designed to leverage local attention mechanisms, reminiscent of those found in Graph Neural Networks (GNNs). This design choice enables them to optimize message passing effectively and capture intricate local attentions within graphs (Rong et al., 2021; Thakoor et al., 2022; Bo et al., 2023), often neglecting the global attention capabilities inherent in the Transformer model. However, in contrast, certain methodologies place an

excessive emphasis on the global attention capabilities inherent in the Transformer model (Chen et al., 2023; Ma et al., 2023; Zhao et al., 2023). While these approachs aims to capture global dependencies within the graph, it often undervalues the critical role of local attention mechanisms in understanding the nuanced context of graph structures.

In this work, we initially propose a novel Message Passing Transformer with Considered modifications to the original Transformer, significantly enhancing its performance on graph tasks through a more sophisticated handling of local attention.. Building upon this, we further develop a Dual-Block Graph Transformer that effectively integrates both local and global attention mechanisms. This architecture comprises two distinct blocks: the first, based on our Message Passing Transformer, is designed to emulate local attention akin to Message Passing GNNs, and the second encapsulates the global attention mechanism, mirroring the original Transformer model. This dual-block design enables our model to capture both fine-grained local and high-level global interactions in graph tasks, leading to a more comprehensive and robust graph representation. Potential future work could explore different ways to combine the outputs of the multiple blocks, or investigate the use of diverse attention mechanisms for each block.

## 2 RELATED WORKS

### 2.1 GRAPH NEURAL NETWORKS

Graph Neural Networks (GNNs) have emerged as a powerful tool for processing and analyzing graph-structured data, leading to a surge of research and advancements in this field. GNNs have shown their versatility by being applicable across various domains, such as social network analysis, biological network interpretation, and many more. This wide range of applications is a testament to their ability to capture and process the complex relationships inherent in graph data (Kipf & Welling, 2016). A significant development in this area is the Graph Attention Network (GAT) (Veličković et al., 2018). GATs leverage a local attention mechanism to optimize message passing within graph structures, allowing them to focus on the most relevant nodes and edges during computation. This approach has proven effective in many tasks. However, traditional GNNs (Zhang et al., 2021), tend to focus primarily on local node interactions. This local focus often results in these models overlooking the global context of the graph. This limitation can hinder their effectiveness in tasks that require a broader understanding of the entire graph structure.

### 2.2 GRAPH TRANSFORMERS

In parallel to the developments in GNNs, the Transformer model has made significant strides in the field of natural language processing. Introduced by Vaswani et al. (2017), the Transformer model utilizes a self-attention mechanism that allows it to weigh the importance of different elements in a sequence. This mechanism enables the model to capture complex patterns and long-range dependencies in sequence-based tasks, making it particularly effective for tasks like machine translation and text summarization.

However, applying the Transformer model to graph-based tasks presents a unique set of challenges (Mialon et al., 2021), primarily due to the inherent differences between sequences and graphs. Unlike sequences, graphs do not have a fixed order of elements, which complicates the direct application of the Transformer's self-attention mechanism (Ying et al., 2021). Previous work, Dwivedi & Bresson (2021) use transformer to capture neighbour attention of graph, Rong et al. (2021) try to emerge Message-passing module into transformer as embedding, Ying et al. (2021) Ma et al. (2023) use structure information as bias in transformer for graph tasks, although these methodologies excel at capturing local graph structures, they frequently omit the potential of the Transformer model's inherent global attention capabilities.

Contrarily, there exist methodologies that place an excessive emphasis on the global attention capabilities inherent in the Transformer model Zhu et al. (2023) preprocess graph data as a hierarchical sequence for the Transformer, Zhao et al. (2023) explore the deep of transformer layers, Jain et al. (2021) incorporates the Transformer as a post-GNN module to capture global attention of graph features, Chen et al. (2023) preprocess the node features as sequence for the transformer. These approaches, while aiming to capture global dependencies within the graph, often underplay the pivotal role of local attention mechanisms. The nuanced context of graph structures necessitates a balanced

focus on local attention mechanisms, a facet frequently undervalued in these methodologies. This imbalance underscores the need for a model that effectively integrates both local and global attention mechanisms.

To address this, we introduce an impactful modification to the original Transformer model in our design. We maintain both local and global attention mechanisms but arrange them in a parallel configuration. This design aims to more effectively capture the complex relationships present in graph data.

## 3 BACKGROUND

A graph $\mathcal{G}$ is a mathematical structure used to model pairwise relations between objects. It is defined as an ordered pair $\mathcal{G} = (\mathcal{V}, \mathcal{E})$, where $\mathcal{V}$ is the set of vertices or nodes, and $\mathcal{E}$ is the set of edges. Each edge is a 2-element subset of $\mathcal{V}$.

The adjacency matrix is a square matrix used to represent a finite graph. The elements of the matrix indicate whether pairs of vertices are adjacent or not in the graph. For a graph with $N$ nodes, the adjacency matrix $\mathbf{A}$ is an $N \times N$ matrix where the entry $A_{ij}$ is 1 if there is an edge from node $i$ to node $j$, and 0 otherwise.

### 3.1 TRANSFORMERS

The Transformer model (Vaswani et al. (2017)) is based on the self-attention mechanism, which is often described in terms of query, key, and value (QKV) vectors. Given a sequence of input vectors $\mathbf{X} = (\mathbf{x}_1, \mathbf{x}_2, ..., \mathbf{x}_n)$ $in \mathbb{R}^{n \times d}$, the self-attention mechanism first computes the QKV vectors as follows:

$$\mathbf{Q} = \mathbf{X}\mathbf{W}_Q, \quad \mathbf{K} = \mathbf{X}\mathbf{W}_K, \quad \mathbf{V} = \mathbf{X}\mathbf{W}_V \tag{1}$$

Where $\mathbf{W}_Q, \mathbf{W}_K$, and $\mathbf{W}_V \in \mathbb{R}^{d \times d_k}$ are the weight matrices to be learned. Finally the self-attention mechanism then computes the output matrix $\mathbf{Y}$ as follows:

$$\mathbf{Y} = \text{softmax}\left(\frac{\mathbf{Q}\mathbf{K}^\top}{\sqrt{d_k}}\right)\mathbf{V} \tag{2}$$

Here, $d_k$ is the dimensionality of the key vectors, and the division by $\sqrt{d_k}$ is a scaling factor that was found to improve performance in the original Transformer paper.

### 3.2 GRAPH NEURAL NETWORKS

Graph Neural Networks (GNNs) operate on graph-structured data represented as $\mathcal{G} = (\mathcal{V}, \mathcal{E})$, where $\mathcal{V} \in \mathbb{R}^{N \times k}$ is the set of nodes and $\mathcal{E}$ is the set of edges. Each node $v_i \in \mathcal{V}$ is associated with a feature vector $\mathbf{x}_i$. The goal of GNNs is to learn a function $f : \mathcal{V} \to \mathbb{R}^d$ that maps nodes to feature vectors, capturing the graph structure.

A common approach in GNNs is to update the feature vector of each node based on its neighbors' feature vectors. For instance, the Graph Convolutional Network (GCN) (Kipf & Welling, 2016) updates the feature vector of node $v_i$ as follows:

$$\mathbf{H}^{(l+1)} = \sigma\left(\tilde{\mathbf{D}}^{-\frac{1}{2}}\tilde{\mathbf{A}}\tilde{\mathbf{D}}^{-\frac{1}{2}}\mathbf{H}^{(l)}\mathbf{W}^{(l)}\right) \tag{3}$$

where $\mathbf{A} \in \mathbb{R}^{N \times N}$ is the adjacency matrix of the graph, $\tilde{\mathbf{A}} = \mathbf{A} + \mathbf{I}$ is the adjacency matrix with added self-connections, $\mathbf{I}$ is the identity matrix, $\tilde{\mathbf{D}}$ is the diagonal node degree matrix of $\tilde{\mathbf{A}}$, $\mathbf{H}^{(l)}$ is the matrix of node features at layer $l$, $\mathbf{W}^{(l)}$ is a learnable weight matrix at layer $l$, and $\sigma$ is a non-linear activation function.

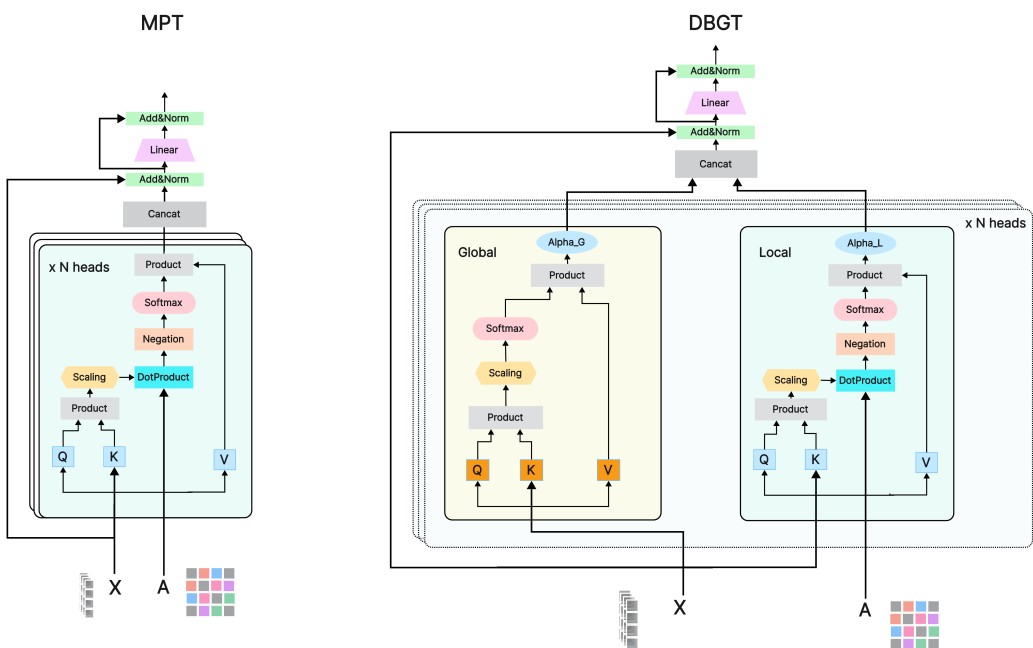

Figure 1: The model of Message-Passing Transformer (MPT) and Dual-block Graph Transformer (DBGT). Left: Message-Passing Transformer (MPT) fully leverages local attention. Right: DBGT model processes feature embeddings and an adjacency matrix through two blocks: a Global-block for global attention, and a Local-block (MPT block) for local attention. The outputs are concatenated and passed through a linear feed-forward network with a residual connection.

## 4 OUR APPROACH

Recognizing the challenges inherent in applying the Transformer model to graph-structured data, where some methodologies tend to overly concentrate on either local or global attention mechanisms, we propose a novel, two-pronged solution. Initially, our approach focuses on the development of a graph transformer that fully leverages local attention. This model, which we term the Message-Passing Transformer (MPT) as shown in 1, is designed to emulate the entire graph message-passing process in whole graph attention networks (GAT), effectively serving as a local attention Graph Transformer. Subsequently, we expand this model by integrating a fully global attention block as Dual-Block Graph Transformer (DBGT), thereby creating a comprehensive system that balances both local and global attention mechanisms. This progressive approach allows us to harness the strengths of both attention mechanisms, potentially leading to more robust and effective graph-structured data processing.

### 4.1 MESSAGE PASSING TRANSFORMER

In the Message-Passing Transformer (MPT) model, the attention mechanism is first applied. The input features $\mathbf{X}$ of each node are transformed into query, key, and value vectors by $\mathbf{W}_Q$, $\mathbf{W}_K$, and $\mathbf{W}_V$, the attention mechanism then computes the output matrix $\mathbf{M}$ as follows:

$$\mathbf{M} = \frac{\mathbf{Q}\mathbf{K}^\top}{\sqrt{d_k}} \qquad (4)$$

where $\mathbf{M}$ is the attention matrix and $d_k$ is the dimensionality of the key vectors. The division by $\sqrt{d_k}$ is a scaling factor that was found to improve performance in the original Transformer paper.

In order to ensure that the attention mechanism does not consider non-adjacent nodes, we apply a Neglation operation to the attention scores before the softmax operation. Specifically, we replace

the scores corresponding to non-adjacent node pairs (i.e., pairs of nodes for which the adjacency matrix entry is zero) with a large negative value (i.e., -9e15). This effectively forces the softmax of these scores to be near zero, ensuring that the attention mechanism does not consider non-adjacent nodes. Mathematically, this operation can be represented as follows:

$$\mathbf{M}' = \text{Negation}(\mathbf{M}) : \text{where}\{\text{if } \mathbf{M_{ij}} == 0, \mathbf{M_{ij}} = -\infty, \ \mathbf{M_{ij}} = \mathbf{M_{ij}} \text{ otherwise}\} \tag{5}$$

The adjacency matrix $\mathbf{A}$ is then used to compute a weighted sum of the value vectors which make the message passing of local attention:

$$\mathbf{Z} = \text{softmax}(\mathbf{A} \odot \mathbf{M}')\mathbf{V} \tag{6}$$

Where $\odot$ represents the dot product, its application in conjunction with the adjacency matrix $\mathbf{A}$ serves a crucial role. Specifically, $\mathbf{A}$ functions akin to a mask, selectively highlighting neighbor nodes while obscuring all others. Upon multiplication with $\mathbf{V}$, this mechanism effectively restricts the self-attention mechanism to focus solely on each node's neighbouring local attention. The output matrix $\mathbf{Z}$ is then passed through a linear feed-forward layer:

$$\mathbf{H} = \text{ReLU}(\mathbf{W}_1 \mathbf{Z} + \mathbf{b}_1) \tag{7}$$

$\mathbf{W}_1$ and $\mathbf{b}_1$ are the parameters of the feed-forward network. The output of the feed-forward network is then combined with the original features through a residual connection:

$$\mathbf{Y} = \mathbf{H} + \mathbf{X} \tag{8}$$

This operation ensures that the original features are preserved while being refined by the aggregated information. The combined features are then normalized to stabilize the learning process. The layer-wise operation is repeated in the following layer, allowing the model to capture higher-order neighborhood information. In other words, the model can aggregate information from neighbors of neighbors, and so on, effectively capturing a larger local graph structure around each node with each layer.

## 4.2 Dual-Block Graph Transformer (DBGT)

Having obtained the local attention via our MPT, the subsequent challenge lies in effectively dealing with the global attention within the graph transformer in a manner that coheres with our proposed local attention model, MPT. Rather than utilizing the transformer as an isolated module, we adapt it to function synergistically with the local attention in MPT as a dual-block within each head. This approach not only ensures a seamless integration of local and global attention mechanisms but also enhances the overall effectiveness of the model.

To this end, We propose the Dual-Block Graph Transformer (DBGT) as shown in Figure 1, a novel extension of our MPT. The DBGT model integrates both local and global attention mechanisms by incorporating two distinct 'blocks': a Graph Attention block and a Transformer block. The Graph Attention block focuses on local attention, capturing the local graph structure around each node. It operates similarly to the attention mechanism in the MPT model, computing a weighted sum of the value vectors of the neighboring nodes for each node:

$$\mathbf{Z}^{\text{L}} = \text{softmax}(\mathbf{A} \odot \text{Negation}(\frac{\mathbf{Q_L}\mathbf{K_L}^\top}{\sqrt{d_k}}))\mathbf{V_L} \tag{9}$$

where $\mathbf{A}$ is the adjacency matrix of the graph, and negation is the operation of equation (5). The Transformer block, on the other hand, captures global attention, considering all nodes in the graph when computing the attention weights:

$$\mathbf{Z}^{\text{G}} = \text{softmax}\left(\frac{\mathbf{Q_G}\mathbf{K_G}^\top}{\sqrt{d_k}}\right)\mathbf{V_G} \tag{10}$$

The outputs of the two blocks are then concatenated to form the final output of the DBGT model:

$$\mathbf{Z} = \text{Concat}(\alpha_L \mathbf{Z}^{\text{L}}, \alpha_G \mathbf{Z}^{\text{G}}) \tag{11}$$

where $\alpha_L$ and $\alpha_G$ are two learned parameters, and we set $\alpha_L = (\mathbf{1} - \alpha_{\mathbf{G}})$. Furthermore, we extend the DBGT model to have $N$ heads of Dual-blocks, allowing the model to capture a richer set of local and global dependencies in the graph. This makes the DBGT model a versatile tool for graph representation learning, capable of focusing on both local and global graph structures.

## 5 EXPERIMENTS

To validate the effectiveness of our proposed models, Message-Passing Graph Transformer (MPT) and Dual-Block Graph Transformer (DBGT), we conduct extensive experiments on both node classification and graph classification tasks. These tasks serve as benchmarks in the field of graph-structured data analysis and allow us to compare the performance of our models with existing state-of-the-art methods of GNNs and Graph Transformers.

### 5.1 DATASETS

For node classification tasks, we employ a variety of widely-used datasets. The Cora dataset is a citation network where nodes represent scientific papers and edges represent citation relationships (McCallum et al., 2000). The WikiCS dataset is a network of Wikipedia articles about computer science, with nodes representing articles and edges indicating hyperlinks between them (Zhang et al., 2020). The Actor dataset is a co-occurrence network of actors, with nodes representing actors and edges indicating that two actors appeared in the same movie (State et al., 2014). The Chameleon dataset is a hyperlinked document dataset, where nodes represent documents and edges represent hyperlinks (Rozemberczki et al., 2020). Lastly, the Wisconsin, Texas, and Cornell datasets are web page datasets, where nodes represent web pages and edges represent hyperlinks (Yang et al., 2015). Each of these datasets offers a unique context for evaluating our models, providing a comprehensive assessment of their performance across different domains and structures.

We utilize a variety of benchmark datasets for graph classification tasks. MUTAG predicts the mutagenicity of nitroaromatic compounds (Debnath et al., 1991). PROTEINS classifies proteins into enzymes or non-enzymes based on amino acid proximity (Dobson & Doig, 2003). NCI1 predicts the anti-cancer activity of chemical compounds (Wale et al., 2008). PTC determines the carcinogenicity of tested compounds (Toivonen et al., 2003). These datasets allow a thorough evaluation of our model's versatility in handling diverse graph classification tasks.

### 5.2 BASELINE

In our node classification and graph classification experiments, we compare our proposed models against a diverse set of baselines, which fall into two main categories: Graph Neural Networks (GNNs) and Graph Transformers.

**Graph Neural Networks** We selected a variety of GNN models that have shown strong performance on graph-based tasks including heomophlic graph tasks and heterophilic graph tasks. These include Graph Convolutional Networks (GCN) (Kipf & Welling, 2016), Graph Attention Networks (GAT) (Veličković et al., 2018), GIN (Xu et al., 2019), SAGE (Hamilton et al., 2017), GPR-GNN (Chien et al., 2021), Shadow-GCN (Zeng et al., 2021), JacobiConv (Wang & Zhang, 2022), BGRL (Thakoor et al., 2022), and GREET (Liu et al., 2023). These models employ different mechanisms for information propagation and aggregation, providing a comprehensive comparison for our proposed models.

**Graph Transformers** We also compare our models against several Graph Transformer models, which integrate the self-attention mechanism of Transformers with the propagation mechanism of GNNs. These include the basic Transformer model (Vaswani et al., 2017), Graphormer (Ying et al., 2021), GT (Yun et al., 2019), Graphi (Mialon et al., 2021), SAT (Chen et al., 2022), GRPE (Park et al., 2022), SpecFormer (Bo et al., 2023), Deepgraph (Zhao et al., 2023), NAGphormer (Chen et al.,

Table 1: Test accuracy results for the Graph node classification task. The highest and second highest results are highlighted in bold and underlined, respectively.

|  | Cora | Wikics | photo | Actor | Cham. | Wis. | Texas | Cornell |
|---|---|---|---|---|---|---|---|---|
| GCN | 80.04 | 76.21 | 92.35 | 31.06 | 38.11 | 57.03 | 58.43 | 52.15 |
| GAT | 83.28 | 77.37 | 92.64 | 28.16 | 43.71 | 55.76 | 57.96 | 50.58 |
| GPR-GNN | 74.51 | 79.00 | 92.42 | 31.06 | 60.72 | 78.26 | 79.28 | 78.40 |
| Shadow-GCN | 81.69 | 79.11 | 91.68 | 32.33 | 56.89 | 64.05 | 61.80 | 62.59 |
| Jcobiconv (Wang & Zhang, 2022) | 81.70 | 76.28 | 92.26 | 31.26 | 63.89 | 76.64 | 66.58 | 65.00 |
| BGRL (Thakoor et al., 2022) | 81.38 | 78.89 | 93.24 | 30.89 | 48.08 | 52.59 | 60.00 | 57.53 |
| GREET (Liu et al., 2023) | **83.81** | 80.68 | 92.85 | 36.20 | 63.70 | 81.90 | 87.00 | 75.10 |
| GT | 71.54 | 73.41 | 91.16 | 30.32 | 52.67 | 72.86 | 71.85 | 71.16 |
| Graphormer | 70.47 | 68.08 | 89.67 | 29.33 | 56.92 | 71.60 | 74.25 | 72.04 |
| Graphi | 78.44 | 74.70 | 90.70 | 32.68 | 59.99 | 68.21 | 68.43 | 66.91 |
| SAT (Chen et al., 2022) | 81.51 | 75.87 | 91.28 | 34.67 | 64.66 | 75.20 | 74.81 | 75.02 |
| GRPE (Park et al., 2022) | 80.12 | 75.28 | 90.41 | 33.51 | 64.71 | 76.60 | 74.07 | 73.17 |
| SpecFormer (Bo et al., 2023) | 80.46 | 78.99 | 92.36 | 34.49 | 64.74 | 77.06 | 78.06 | 76.45 |
| Deepgraph (Zhao et al., 2023) | 79.49 | 77.52 | 91.22 | 31.48 | 62.65 | 74.09 | 75.30 | 71.28 |
| NAGphormer (Chen et al., 2023) | 80.79 | 78.70 | 90.04 | 34.11 | 59.24 | 76.07 | 76.76 | 75.86 |
| AGT (Ma et al., 2023) | 81.57 | 78.93 | 92.68 | 35.19 | **75.06** | 80.67 | 79.83 | 76.10 |
| **MPT** | 81.71 | **82.90** | 93.86 | 36.63 | 65.53 | 78.63 | 86.49 | 80.82 |
| **DBGT** | 81.01 | 81.07 | **94.12** | **37.14** | 66.36 | **83.88** | **89.19** | **81.89** |

2023) and AGT (Ma et al., 2023). These models represent popular and state-of-the-art approaches in the field of Graph Transformers, serving as a challenging benchmark for our proposed models.

## 5.3 SETTINGS

**Node Classification** For the node classification task, we followed the settings described in the AGT paper (Ma et al., 2023). We conducted 10 independent runs for each dataset using the standard split, with each run consisting of 100 epochs. The final results are reported as the average over these 10 runs. This approach ensures that our results are robust and not dependent on a particular random seed or initial conditions.

**Graph Classification** For the graph classification task, we employed a 10-fold cross-validation strategy for each model. This involves splitting the dataset into 10 equal parts, or 'folds', and then training the model 10 times, each time using 9 folds for training and the remaining fold for testing. The final results are reported as the average over these 10 folds. This strategy provides a robust estimate of the model's performance, as it reduces the variance associated with a single train-test split.All the experiments are running on NVIDIA TITAN RTX 24G.

## 5.4 RESULTS

Table 1 presents the performance comparison of various Graph Neural Networks (GNNs), Graph Transformers, and our proposed models (MPT and DBGT) on multiple datasets for node classification tasks.In the GNNs category, GREET achieves the highest accuracy on the Cora dataset. GREET also shows impressive performance on the Wikics, Actor, and Wisconsin datasets. However, our proposed models, MPT and DBGT, outperform or are competitive with these models across all datasets. In the Graph Transformers category, AGT stands out with the highest accuracy on the Cham dataset. However, our proposed models again demonstrate superior performance. MPT outperforms

Table 2: Test accuracy results for the Graph classification task. The highest and second highest results are highlighted in bold and underlined, respectively

|  | MUTAG | PROTEINS | NCI1 | PTC |
|---|---|---|---|---|
| GCN | 74.6 ± 7.7 | 73.1 ± 3.8 | 76.2 ± 4.1 | 60.4 ± 3.2 |
| SAGE | 74.9 ± 8.7 | 73.4 ± 3.3 | 71.7 ± 3.1 | 58.8 ± 4.2 |
| GIN | 85.4 ± 7.7 | 72.1 ± 5.2 | 78.6 ± 2.9 | **66.4 ± 6.6** |
| GT | 75.5 ± 7.9 | 68.4 ± 3.3 | 67.7 ± 4.3 | 55.6 ± 5.2 |
| Graphormer | 74.4 ± 7.4 | 68.4 ± 3.7 | 64.9 ± 3.7 | 56.7 ± 5.5 |
| SAT (Chen et al., 2022) | 81.4 ± 8.2 | 71.7 ± 3.5 | 70.2 ± 3.5 | 61.3 ± 4.0 |
| GRPE (Park et al., 2022) | 82.5 ± 8.0 | 72.3 ± 3.5 | 74.7 ± 5.6 | 63.1 ± 3.4 |
| SpecFormer (Bo et al., 2023) | 83.7 ± 8.0 | 72.0 ± 4.9 | 76.7 ± 4.2 | 63.2 ± 4.7 |
| Deepgraph (Zhao et al., 2023) | 80.5 ± 8.1 | 70.5 ± 3.8 | 74.4 ± 5.3 | 62.5 ± 4.8 |
| AGT (Ma et al., 2023) | 82.8 ± 8.2 | 73.2 ± 4.5 | 76.1 ± 3.5 | 63.4 ± 4.1 |
| **MPT** | 84.3 ± 8.4 | 75.4 ± 3.1 | 74.5 ± 3.6 | 64.0 ± 3.6 |
| **DBGT** | **88.6 ± 7.2** | **76.9 ± 3.3** | **79.4 ± 2.6** | 65.4 ± 4.8 |

all Graph Transformers on the Wikics and Actor datasets, while DBGT achieves the highest accuracy on the Wikics, Actor, photos, Wisconsin, Texas, and Cornell datasets and the second-highest on the Chameleon dataset.

Table 2 showcases the performance on graph classification tasks.In the GNNs category, GIN achieves the highest accuracy on the MUTAG, NCI1 and PTC dataset. However, our proposed models, MPT and DBGT, outperform or are competitive with these models. In the Graph Transformers category, AGT stands out with the highest accuracy on the PROTEINS, NCI1 and PTC dataset. However, our proposed models again demonstrate superior performance. DBGT achieves the highest accuracy on the MUTAG, PROTEINS and NCI1, and the second-highest on the PTC dataset.

Our proposed Message Passing Transformer (MPT) demonstrates superior performance compared to most Graph Neural Networks (GNNs) and Graph Transformers across several datasets. This indicates the effectiveness of our strategic modifications to the original Transformer model in handling local attention. Furthermore, with the integration of global attention, our Dual-Block Transformer (DBGT) outperforms nearly all existing models, highlighting the strength of a balanced attention mechanism. However, it's worth noting that our approach shows room for improvement on the homophilic dataset Cora, despite performing well on Wikics and Photo datasets. Particularly on heterophilic datasets, our approach shows significant improvement, underscoring the importance of global information in these contexts.

## 5.5 ABLATION STUDY

Table 3 presents the results of an ablation study conducted on our DBGT model, examining the effects of removing either the local or global attention block in each dataset. The study reveals that in homophilic datasets such as Cora and Wikics, the presence of global attention has a negative impact. Conversely, for heterophilic datasets and graph classification tasks, the role of global attention proves to be crucial. When focusing on local attention, which constitutes the main part of graph message passing, the absence of this component leads to a substantial decrease in performance across all graph tasks.

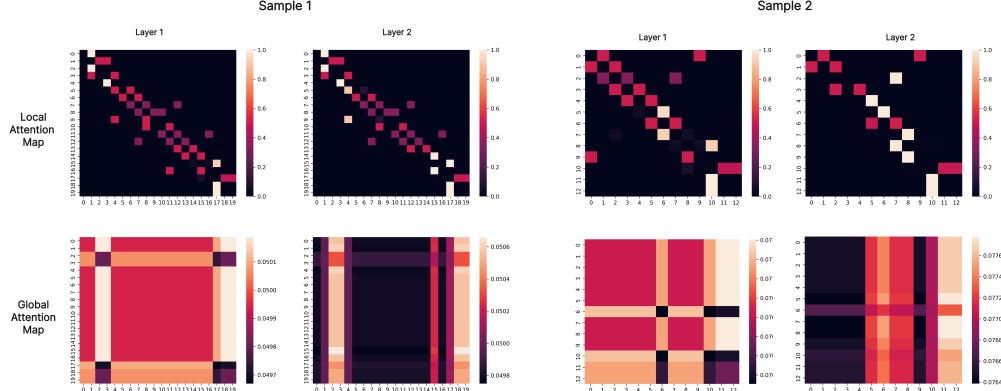

Figure 2: Visualization of two-layer local and global attention maps on two samples from the MU-TAG dataset. The top panel represents the local attention map in each layer, capturing the local attention of each node, while the bottom panel represents the global attention map in each layer, capturing the global attention of each node.

Table 3: Test accuracy results of ablation study for graph tasks. w/o means without.

|  | MUTAG | PROTEINS | NCI1 | PTC | Cora | Wikics | Wis. | Texas |
|---|---|---|---|---|---|---|---|---|
| **DBGT** | 88.6 ± 7.2 | 76.9 ± 3.3 | 79.4 ± 2.6 | 65.4 ± 4.8 | 81.0 ± 0.3 | 81.1 ± 0.2 | 83.9 ± 0.4 | 89.2 ± 0.2 |
| **DBGT w/o Global** | 84.3 ± 8.4 | 75.4 ± 3.1 | 74.5 ± 3.6 | 64.0 ± 3.6 | 81.7 ± 0.2 | 82.9 ± 0.3 | 78.6 ± 3.6 | 86.5 ± 1.6 |
| **DBGT w/o Local** | 67.1 ± 7.5 | 62.6 ± 3.4 | 61.3 ± 3.8 | 52.2 ± 2.9 | 66.5 ± 0.5 | 67.1 ± 0.6 | 59.2 ± 2.1 | 56.5 ± 4.2 |

Figure 2 provides a visualization of the dual block attention map in our DBGT model. We selected two samples from the MUTAG dataset to illustrate the local and global attention mechanisms after a fully trained, two-layer DBGT model. The visualization clearly highlights the differences between the local and global attention maps. This not only demonstrates the effectiveness of the global attention mechanism, which is often neglected by traditional Graph Transformers and GNNs, but also underscores the unique contribution of our DBGT model in addressing this oversight.

## 6 FUTURE WORK

Despite the promising results of our Message-passing Transformer and Dual-Block Graph Transformer, it presents some limitations, particularly in terms of scalability. This issue, inherent to the global attention mechanism, results in a high computational cost, especially for large-scale graphs. Future work should focus on addressing this challenge. Additionally, potential future work could explore different ways to combine the outputs of the multiple blocks, or investigate the use of diverse attention mechanisms for each block. These areas of improvement present exciting opportunities for further advancements in graph-based machine learning.

## 7 CONCLUSION

In conclusion, this work presents an impactful modification to the Transformer architecture that leverages both local and global attention mechanisms, yielding substantial improvements in graph tasks performance. Our findings pave the way for future research in developing more sophisticated and efficient graph transformer models, thereby pushing the frontier of graph-based machine learning.

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
