# OpenReview forum: "Enhancing Graph Tasks with a Dual-Block Graph Transformer: A Synergistic Approach to Local and Global Attention"
_ICLR.cc/2024/Conference — ICLR 2024 Conference Withdrawn Submission_

### Official Review · Reviewer_wt2P · 2023-10-25

**Soundness:** 2 fair
**Presentation:** 3 good
**Contribution:** 2 fair
**Rating:** 5
**Confidence:** 4

**Summary:**

This work addresses the limitations of traditional Transformers in graph tasks. It introduces a novel Message Passing Transformer that enhances the handling of local attention, and a Dual-Block Graph Transformer that integrates local and global attention mechanisms. The proposed models outperform both GNN and Graph Transformer models in node and graph classification tasks, without the need for complex positional encoding strategies.

**Strengths:**

The paper is well-written and easy to follow.

The proposed DBGT (Dual-Block Graph Transformer) is both simple and effective, demonstrating competitive performance across multiple datasets.

**Weaknesses:**

The paper may lack novelty as it utilizes commonly used encoding modules and a simple concatenation operation. From my perspective, it does not meet the standards set by ICLR, despite showing promising results.

Another concern is the small size of the datasets used. The authors should consider utilizing larger datasets such as the OGB ones. However, it seems that the proposed method may not be applicable to such datasets, as it employs a standard Transformer that considers all nodes in the graph to encode global information.

**Questions:**

The authors mention that they used a 10-fold cross-validation strategy, which deviates from the common setting. However, the results presented in Table 1 lack variance information. Additionally, the authors have not provided the source code. It is unclear why the source code was not uploaded.

---

### Official Review · Reviewer_GViT · 2023-10-30

**Soundness:** 3 good
**Presentation:** 3 good
**Contribution:** 2 fair
**Rating:** 5
**Confidence:** 4

**Summary:**

This paper proposes a Dual-Block Graph Transformer combining both local and global attention, which comprises two parallel blocks in each head: the message passing Block and the original transformer block. The message passing block designed to calculate local attention, and the original transformer block to calculate the global attention. The experiments made on node classification tasks and graph classification tasks show that the proposed architecture does have a fair advantage.

**Strengths:**

1.This proposed method outperforms traditional GNN and Graph Transformer models.
2.The paper is well organized and easy to read, with clear writing framework.

**Weaknesses:**

1. The experiments are mainly focus on the node classification and graph classification, which maybe not sufficient. It’s more convincing to conduct experiments on other tasks (such as graph regression tasks) and use more datasets (such as the datasets in paper [1])

2. Besides, there is little analysis deep into why the proposed method work better. It would be better to systematically study on why it works well on some datasets and worse on others.

[1] X. Ma, Q. Chen, Y. Wu, and et al. Rethinking Structural Encodings: Adaptive Graph Transformer for Node Classification Task.

**Questions:**

As the Weaknesses mentioned above, the suggestions are given separately. Besides, in the ablation part, the analysis maybe too simple. For instance, it’s advised that there be reasons why global attention hurts performance in certain situations. And the same suggestion for the whole experiment and analysis part, not just report the performance but think deep into why.

---

### Official Review · Reviewer_PHGs · 2023-10-31

**Soundness:** 3 good
**Presentation:** 3 good
**Contribution:** 2 fair
**Rating:** 3
**Confidence:** 5

**Summary:**

This work proposes a new architecture called Dual-Block Graph Transformer (DBGT) to address the limitations of traditional Transformers in graph-based tasks. The architecture balances local and global attention mechanisms by incorporating two blocks inside each attention head: the Message Passing Block (MPT) for local attention and a second block for global attention. The dual-block design enables the model to capture both local and high-level global interactions effectively. Empirical validation shows performance improvement in node and graph classification tasks compared to existing Graph Neural Networks and Graph Transformer models. The model achieves this without requiring complex positional encoding.

**Strengths:**

S1. The paper addresses a common issue in graph transformer models, to include local information in addition to global information. The local information is achieved by incorporating a message-passing block, to restrict attentions to local structures only.

S2. The paper is generally clear, well organized and well written, although there is minimal discussion in relation to relevant prior work (see details in W1).

S3. Experimental results are generally promising, especially on graph classification tasks.

**Weaknesses:**

W1. The novelty is not strong enough. The presented approach combines a MPT and a global block, both of which have been explored in prior work. The main contribution of MPT, appears to have a similar idea as GAT, where the attention is limited to edges, but this has not been discussed in detail in the paper, especially on why MPT can apparently outperform GAT in most cases. By the way, there is also an improved version of GAT, called GATv2 [a], might work better than GAT in this regard.

[a] How Attentive are Graph Attention Networks? https://arxiv.org/abs/2105.14491

W2. The trade-off between local and global information can be further analyzed. For one, the proposed model works seem to be slightly better on graph classification than node classification. Intuitively, global information might be more important to graph classification. The authors author briefly mentioned that global information is more important on heterophily datasets---potentially, deeper analysis can be conducted to investigate the effect of global or local information on different task format (node vs graph classification), and different data characteristics (homophily vs heterophily).

W3. [This is more of a comment rather than weakness.] The authors briefly mentioned the scalability issue in future work. At a high-level, this might be the #1 issue with current graph transformers, which prevents large scale deployment.

Minor typos/presentation issues:

- Intro: propose a novel Message Passing Transformer with **Considered** modifications --> Does not seem to be the right word here.

- Below Eq. 4: Neglation --> Negation

- I'm not sure how to read the where condition in Eq (5).

- 5.3: (1) 'fold' --> should use a pair of inverted comma.
(2) with a single train-test split.All ... --> space after full stop.

- 5.4 last paragraph: it's --> it is (should avoid shorthands in formal writing)

- 6/7 future work and related work can be combined together for a non-survey paper.

**Questions:**

See weaknesses above.

**Details Of Ethics Concerns:**

nil

---

### Official Review · Reviewer_NDKD · 2023-11-01

**Soundness:** 2 fair
**Presentation:** 2 fair
**Contribution:** 1 poor
**Rating:** 3
**Confidence:** 5

**Summary:**

The paper proposes a way to adopt the transformer architecture for graphs. The approach consists of capturing graph interaction on a local and global scale. For local interaction, it uses the proposed Message Passing transformer (MPT) which masks out the non-adjacent nodes in the attention matrix (M) along with using the Adjacency matrix (A) followed by the softmax operation. For global interaction, it just considers all graph nodes as tokens and applies the regular transformer model. It then concatenates the local and global representation of the node to obtain the final representation. The final model termed DBGT, is evaluated on node and graph level tasks.

**Strengths:**

- The paper is easy to read and follow.
- The paper focuses on an important problem which is prevalent in the graph learning community which is to capture global and local interactions effectively.

**Weaknesses:**

- The local module only applies the transformer module after the aggregation step for capturing local interactions, which is not proven why it should be better than any standard message-passing framework which already exists.
- The motivation is not strong enough, though the problem is important it is not conveyed clearly in which aspect MPT/DBGT improve on the existing baselines.
- The scalability of the model is also an issue, though the authors point out it is a weakness of the current architecture, the baselines of the paper are scalable as compared to MPT.

**Questions:**

- Given in Figure 2 of the attention visualisations, why are the global attention maps not in the range 0 to 1, it looks like from the visualizations all the nodes get the same attention values because the scale on the right is so small (0.0506 to 0.00498)
- For the Negation operator: This effectively forces the softmax of these scores to be near zero, ensuring that the attention mechanism does not consider non-adjacent nodes. Then why is it that in equation 6 the adjacency matrix is again used as a mask? How are these both different?
- How does equation 6 capture the local neighbourhood interaction can the authors explain it better, it is not evident from the equation why this is the case.
- Is the experimental setup of the paper similar to that of baseline AGT, if yes, why are not all the datasets included for evaluation which is used in AGT (Ogbn-arXiv Arxiv-year, Pokec)?
- In dataset Wisconsin for AGT the value is 83.20 ± 4.10 whereas in the current paper, it is given as 80.67, any reason?
- How the local attention different from the sparse attention mentioned in the paper Rethinking Graph Transformers with Spectral Attention (NeurIPS 2021)

Typos:
Neglation -> Negation

---

### Official Review · Reviewer_5j2q · 2023-11-08

**Soundness:** 2 fair
**Presentation:** 3 good
**Contribution:** 2 fair
**Rating:** 3
**Confidence:** 4

**Summary:**

This work propose a dual block graph transformer by combining local dan global attention. More specifically, global attention extends the receptive field to the complete set of nodes. Experimental results on graph classification and node classification tasks are conducted on multiple datasets to show its performance. The presentation

**Strengths:**

Strengths:
1. This work proposes to combine the local and global attention as the key part of transformer layer.
2. Experiments are conducted on multiple benchmark datasets.

**Weaknesses:**

Weaknesses:
1. The novelty is limited to just combine existing idea. Both local and global attention are not new idea.
2. The computation complexity is overlooked and not optimized. In Equation (10), it's just dense attention over the complete graph, which can cause huge computation cost for large graphs.
3. The experiment performance does not support the generation capability of the proposed attention layer for different tasks. For example, the node classification performance does not beat the classical method like GAT, which utilizes the local attention layer.

**Questions:**

Please refer to the weakness mentioned above.